# Farm Machine Use and Pesticide Expenditure in Maize Production: Health and Environment Implications

**DOI:** 10.3390/ijerph16101808

**Published:** 2019-05-21

**Authors:** Jing Zhang, Jianhua Wang, Xiaoshi Zhou

**Affiliations:** 1School of Public Affairs, Zhejiang University, Hangzhou 310058, China; Jingzhang199002@126.com; 2School of Business, Jiangnan University, Wuxi 214122, China; 3Food Safety Research Base of Jiangsu Province, Jiangnan University, Wuxi 214122, China; 4College of Economics & Management, Huazhong Agricultural University, Wuhan 430070, China; Xiaoshi.Zhou@outlook.com

**Keywords:** farm machine use, pesticide expenditure, ESR, China

## Abstract

Although chemical pesticide use has increased agricultural productivity, it has caused adverse effects on human health and the environment. For example, pesticide exposure may result in the incidence of a human health condition (e.g., heart disease, immune disorders, cancer, and damaged skin) and it can pollute air, water, and soil conditions and damage biodiversity. Mitigating the negative externalities associated with pesticide use is essential to improve human health and environmental performance. In this study, we are trying to explore whether farm machine use reduces pesticide expenditure by analyzing farm household survey data collected from 493 maize farmers in China. An endogenous switching regression model is employed to address the sample selection bias issue associated with voluntary farm machine use. The empirical results reveal that farm machine use exerts a negative and statistically significant impact on pesticide expenditure. The findings highlight the important role of farm machines in helping reduce pesticide expenditure, which is, in turn, beneficial for improving human health conditions and environmental performance.

## 1. Introduction

Modern agriculture depends heavily on pesticide use, which has successfully increased productivity but led to increasing concerns regarding the environment and human health [1,2,3,4,5]. For example, Richter [6] estimated that about 26 million pesticide poisoning cases resulted in 220 thousand deaths per year worldwide. China is the world’s largest pesticide user and experienced a dramatic increase in pesticide use. In particular, the quantity of pesticide used was dramatically increased over time, from 76.53 thousand tons in 1990 to 176.30 thousand tons in 2014 (see Figure A1 in Appendix A). The booming demand for pesticide in China expanded the share of total world pesticide use and maintained the proportion above 40% over time after 1995. Therefore, a zero-growth action plan for pesticide use has been proposed in China to mitigate agricultural environmental pollution and promote environmental sustainability [7].

Chemical pesticide use contributes to increased agricultural production and improved product quality, but overuse or abuse of pesticides has been related to negative externalities such as human health effects and external ecological effects [3,8,9,10,11,12]. For example, Lai [3] found that a 10% increase in pesticide use in rice production can result in 2.13 million dollars in medical costs in China. Therefore, various policies and measures such as maximum pesticide residue limit (MRL) and integrated pest management (IPM) practices, which target food safety and conservation production, have been released in many developing and developed countries. For example, Midingoyi et al. [13] found that the IPM-adopting farmers had used less insecticide in mango production in Kenya. In their analysis of Dutch farmers, Skevas et al. [14] assessed the effectiveness of different economic policies including taxes, price penalties, and subsidies and quotas, and found that quotas of pesticides were a more effective method for reducing pesticide use.

In addition to the deleterious effects of pesticide use on the environment and health of consumers, the negative effects of chemical pesticides on sustainable development of agriculture have also been well documented [3,4,8,14,15]. Despite these negative externalities of pesticide use, farmers continue to use a high quantity of pesticide, and policies in developing countries do not address the risk of pesticides adequately [16,17].

Farm machines play an important role in sustainable and conservation agriculture development in emerging and developing countries [18,19,20,21]. A considerable body of literature has analyzed the impact of the use of farm machines on agricultural production [21,22,23,24,25]. For example, Benin [18] revealed that agricultural mechanization services used by farming households significantly increase farm yields in Ghana. In their analysis of northern Bangladesh, Rahman et al. [25] showed a positive association between farm machine use and wheat yields. However, there are no previous studies, to the best of our knowledge, that have examined the relationship between farm machine use and pesticide expenditure. It remains unclear whether farm machine use has increased or decreased pesticide expenditure, especially in the presence of self-selection bias associated with voluntary farm machine use.

The impact of farm machine use on pesticide expenditure may be negative or positive. On the one hand, farm machine use can help improve pesticide spraying efficiency, which induces smallholder farmers to spray less. In this case, farm machine use may reduce pesticide expenditure. On the other hand, farm machine use can increase the amount of pesticide use by eliminating the constraints and obligations of pesticide use (e.g., manually spraying skills and physical conditions), resulting in a positive relationship between farm machine use and pesticide expenditure. In addition, farmers themselves decide whether or not to use farm machines for pesticide application (i.e. a self-selection process). In particular, farmers’ farm machine use decisions are likely to be influenced by both observed factors (e.g., age, gender, household size) and unobserved characteristics (e.g., farmers’ innate abilities and managerial skills) that may be correlated to the outcome of interest (pesticide expenditure in our case) [19,26]. This fact leads to a sample selection and endogeneity issue, which needs to be addressed in order to obtain an unbiased and consistent estimation of the treatment effect of farm machine use on pesticide expenditure.

The primary objective of this study is to analyze the impact of farm machine use on pesticide expenditure, utilizing data observed at the farm-level in rural China. This study aims to contribute to the growing literature on the role of farm machines in the development of sustainable and conservation agriculture from two aspects. First, we investigate the determinants of farm machine use for pesticide applications by using recently collected survey data from 493 smallholder maize farmers in rural China. The existing studies have so far paid more attention to the total machine power or machine use intensity in their efforts to analyze their determinants. However, studies are almost completely absent on farm machine use at a specific production stage such as pesticide application [20]. Second, we take the possible endogeneity of farm machine use into account, in particular stemming from the self-selection process, in our estimated econometric models. To achieve this, we use an endogenous switching regression (ESR) model to address the potential selection-bias issue arising from both observed and unobserved factors [27]. To date, no previous studies have accounted for the impact of farm machine use on pesticide expenditure.

The remainder of this paper unfolds as follows. The data and descriptive statistics are presented in the next section. Section 3 presents the empirical specification. This is followed by presentation of the estimated results and discussion in Section 4, and Section 5 concludes.

## 2. Empirical Specification

### 2.1. The Decision of Farm Machine Use

We assume that farmers are rational decision-makers and account for the potential net returns obtained from using or not using farm machines in maize production. Let the net return obtained from using the farm machines be $A_{U}^{*}$, and that received from not using them be $A_{N}^{*}$. A rational farmer may choose to use the farm machines if the net return difference ($A_{i}^{*}$) between using and not using is positive, $A_{i}^{*}=A_{U}^{*}-A_{N}^{*}>0$. However, $A_{i}^{*}$ is unobserved, which can be expressed as a function of observable factors by a latent variable model as follows:(1)$$A_{i}^{*}=X_{i}\beta +\mu_{i},\;A_{i}=1\;if\;A_{i}^{*}>0$$
where $A_{i}$ is an observed dichotomous indicator variable which represents the farm household i’s farm machine use status, i.e., $A_{i}=1$ for household i who use farm machines to apply pesticides, and 0 otherwise. $X_{i}$ refers to a vector of household and farm level characteristics (e.g., age, gender, education, off-farm work participation, household size, farm size); β represents the corresponding parameters to be estimated; and $\mu_{i}$ is a random error term with a mean of zero and variance $\sigma_{\mu_{i}}^{2}$. The probability of using farm machines can be expressed as:(2)$$\mathrm{Pr}\left(A_{i}=1\right)=\mathrm{Pr}\left(A_{i}^{*}>0\right)=\mathrm{Pr}\left(\mu_{i}>-X_{i}\beta \right)=1-F\left(-X_{i}\beta \right)$$
where F(·) is the cumulative distribution function for $\mu_{i}$.

### 2.2. Impact Evaluation and Selection Bias

The present study aims to investigate the impact of farm machine use on pesticide expenditure. Given that pesticide expenditure is a linear function of a vector of household and farm level characteristics, the outcome variable can be expressed as:(3)$$Y_{i}=Z_{i}\eta +A_{i}\gamma +\varepsilon_{i}$$
where $Y_{i}$ refers to the pesticide expenditure variable; $Z_{i}$ represents the household and farm-level characteristics (e.g., age, gender, education, household size, and farm size); $A_{i}$ refers to the previously described indicator of farm machine use on spraying pesticide; $\varepsilon_{i}$ is a random error term; and η and γ are vectors of parameters to be estimated.

Unobserved characteristics may also affect farmers’ farm machine use decision and the pesticide expenditure, and this results in a potential endogeneity issue of farm machine use variables. For example, farmers who have higher farm managerial ability may be more likely to use farm machines to spray their pesticides. Since farm managerial ability is not observed, this may result in a non-zero correlation coefficient of the error terms, i.e., ρ=corr(ε,μ)≠0. In this case, estimating Equation (3) using an ordinary least square (OLS) regression model would produce a biased estimate with respect to the effects of farm machine use on pesticide expenditure.

The propensity score matching (PSM) approach is commonly used in impact evaluation programs, in particular when self-selection occurs [28,29,30]. However, one of the major drawbacks of the PSM approach is that it accounts only for the selection bias resulting from observed factors. To simultaneously estimate the determinants and the impacts of farm machine use on pesticide expenditure, while accounting for both observed and unobserved factors in an efficient manner, an endogenous switching regression (ESR) model developed by Lokshin and Sajaia [27] was employed in the present study. The ESR model has been applied in previous studies [31,32,33]. For example, Kabunga et al. [33] employed the ESR model to examine the yield effects of tissue culture banana technology adoption in Kenya. Ma and Abdulai [31] also applied the ESR model to investigate the impact of cooperative membership on farm performance in China, and they found cooperative membership contributes to the increase of apple yields, farm net returns, and household income.

### 2.3. Endogenous Switching Regression Model

A two-stage estimation procedure is estimated simultaneously in the ESR model framework. In the first stage, the determinants influencing farm machine use are estimated based on the selection Equation (1). In the second stage, the impacts of farm machine use on the outcome variable are specified for tow regimes of farm machine users and nonusers as:(4a)$$\mathrm{Regime}\;1\;(\mathrm{farm}\;\mathrm{machine}\;\mathrm{users}):\;Y_{iU}=Z_{i}\beta_{iU}+\varepsilon_{iU}\;if\;A_{i}=1$$
(4b)$$\mathrm{Regime}\;2\;(\mathrm{farm}\;\mathrm{machine}\;\mathrm{nonusers}):\;Y_{iN}=Z_{i}\beta_{iN}+\varepsilon_{iN}\;if\;A_{i}=0$$
where $Y_{iU}$ and $Y_{iN}$ are pesticide expenditure variables for farm machine users and nonusers, respectively; $Z_{i}$ represents the household and farm level characteristics as defined previously; $\beta_{U}$ and $\beta_{N}$ refers to the corresponding parameters to be estimated; and $\varepsilon_{iU}$ and $\varepsilon_{iN}$ are the random error terms.

It is worth noting that the variables $Z_{i}$ in Equation (4a,b) and $X_{i}$ in Equation (1) are allowed to overlap. However, proper identification requires at least one variable in $X_{i}$ that does not appear in $Z_{i}$. Thus, the selection equation is estimated using the same variables in the outcome equation in addition to at least an identifying instrument. A valid instrument is expected to influence farm machine use decision but does not affect pesticide expenditure directly. In this study, we follow previous studies and include an instrumental variable representing the rate of farm machine use in spraying pesticides in households per village as an additional regressor in Equation (1) [34]. The rate of farm machine use to apply pesticides per village is hypothesized to affect farmers’ decision of farm machine use but not the outcome (i.e., pesticide expenditure). This is considered to be a valid and relevant instrument.

The variables in $Z_{i}$ in Equation (4a,b) account for the observed factors. However, the ESR model can be used to address the selection bias arising from unobserved factors within a context of the omitted variable problem. Specifically, Heckman (1979) [35] indicated that the inverse mills ratios or selectivity terms can be calculated after estimating the selection Equation (1) and then included in Equation (4a,b) to address selection bias stemming from unobserved factors. The covariance terms are plugged into Equation (4a,b) to obtain (5a,b) and specified as: (5a)$$Y_{iU}=Z_{i}\beta_{U}+\sigma_{\mu U}\lambda_{U}+\vartheta_{iU}\;if\;A_{i}=1$$
(5b)$$Y_{iN}=Z_{i}\beta_{N}+\sigma_{\mu N}\lambda_{N}+\vartheta_{iN}\;if\;A_{i}=0$$
where $\sigma_{\mu U}$ and $\sigma_{\mu N}$ represent covariance terms between the error terms in the selection and outcome equations; $\lambda_{U}$ and $\lambda_{N}$ are selectivity terms; and $\vartheta_{iU}$ and $\vartheta_{iN}$ are the error terms with conditional zero means.

In addition to estimating the factors that influence farm machine use and pesticide expenditure for users and non-users, the ESR model can also be used to examine the treatment effects of farm machine use on pesticide expenditure. The treatment effects of farm machine use on pesticide expenditure are examined by comparing the expected pesticide expenditure of farm households who use farm machines with the expected outcomes of the counterfactual hypothetical cases where users did not use farm machines. The expected values of the outcome $Y_{i}$ on farm machine users and nonusers can be expressed as in Equation (6a,b): (6a)$$E\left(Y_{iU}|A=1\right)=Z_{i}\beta_{iU}+\sigma_{\mu U}\lambda_{iU}$$
(6b)$$E\left(Y_{iN}|A=0\right)=Z_{i}\beta_{iN}+\sigma_{\mu N}\lambda_{iN}$$

Accordingly, the expected value of the farm machine users, had they chosen not to use farm machines, and of the same nonusers had they chosen to use farm machines is given, respectively, as follows:(7a)$$E\left(Y_{iN}|A=1\right)=Z_{i}\beta_{iN}+\sigma_{\mu N}\lambda_{iU}$$
(7b)$$E\left(Y_{iU}|A=0\right)=Z_{i}\beta_{iU}+\sigma_{\mu U}\lambda_{iN}$$

A change in the outcome due to farm machine use can be termed as the average treatment effects on the treated (ATT) and the average treatment effects on the untreated (ATU). Following Lokshin and Sajaia [27] and Ma et al. [32], the ATT and ATU can be calculated as follows:(8a)$$ATT=E\left(Y_{iU}|C=1\right)-E\left(Y_{iN}|C=1\right)=Z_{i}\left(\beta_{iU}-\beta_{iN}\right)+\lambda_{iU}\left(\sigma_{\mu U}-\sigma_{\mu N}\right)$$
(8b)$$ATU=E\left(Y_{iU}|C=0\right)-E\left(Y_{iN}|C=0\right)=Z_{i}\left(\beta_{iU}-\beta_{iN}\right)+\lambda_{iN}\left(\sigma_{\mu U}-\sigma_{\mu N}\right)$$
where $\;\sigma_{\mu U},\;\sigma_{\mu N},\;\lambda_{iU},$ and $\lambda_{iN}$ are defined previously.

## 3. Data and Descriptive Statistics

The data used in this study were from a randomized questionnaire survey of Chinese rural households in January 2017, which refers to the production year 2016. The survey was carried out by a multistage random sampling technique. In the first stage, we randomly selected three provinces from western, central, and eastern China, respectively, including Gansu, Henan, and Shandong. The three regions were taken into account because they were different in terms of economic development levels as well as geographic characteristics. In 2016, the cultivated areas of maize in Gansu, Henan, and Shandong were 1.00, 3.31, and 3.21 million hectares, respectively, accounting for about 20.5% of China’s total maize area [36]. In the second stage, one city within each selected province was randomly selected. In particular, Heze city in Shandong, Sanmenxia city in Henan, and Dingxi city in Gansu were randomly selected. In the third stage, in each city we randomly selected three villages. Finally, around 45–55 households were interviewed in each village, resulting in 493 samples in total.

Face-to-face interviews were conducted by well-trained enumerators who spoke both Mandarin and local dialects, using a detailed structured questionnaire. The enumerators were hired from local universities in each province. The survey gathered information covering household and farm-level characteristics (e.g., age, gender, education, household size, farm size), farm machine use status in production and postharvest management, maize yields, production inputs (e.g., pesticide, fertilizer and seed), off-farm work participation status, and access to credit institutions.

The definition and summary statistics of the variables used in this study are presented in Table 1. The dependent variable used in the present study is a dichotomous variable that takes the value of 1 if the household used farm machines to spray pesticides, and the value is 0 otherwise. The outcome variable used in this study refers to the pesticide expenditure per mu. It can be observed from Table 1 that about 57.6% of households in the sample used farm machines to spray their pesticide. The average age of the farm household head is around 47 years, and with a mean household member number of about five. The average 3.5 mu of farm size suggests that the majority of households were small-scale farmers in our sample.

The mean difference in the characteristics of farm machine users and nonusers are presented in Table 2. With respect to the expenditure of pesticides, the results show that the average pesticide expenditure for farm machine users was 14.07 Yuan/mu lower than that of nonusers, which is statistically significant for the mean difference. These descriptive comparison findings seem to suggest that farm machine use plays a critical role in pesticide use, and significantly reduce the pesticide expenditure of farm machine users, relative to nonusers.

In addition, farm machine users are more likely to participate in off-farm work relative to their non-user counterparts. Compared with non-users, farm machine users also have larger farm sizes to cultivate crops. On average, the farm machine users are less likely to take risks and have less probability of accessing credit than nonusers. The transportation conditions are more convenient to train/bus stations for farm machine users than for nonusers. Farm machine users are less likely to receive the agricultural subsidies and less satisfied with the extension service provided by the local government compared with nonusers. The farm machine nonusers’ villages have a higher probability of executing the environment improvement project.

Moreover, the number of farm machine users is significantly lower than the number of nonusers in both Gansu and Henan provinces, however, all the samples in Shandong province are farm machine users. Other household and farm level characteristics such as age and education of household head, household size, and extension contact hardly differ between farm machine users and nonusers. However, the findings in Table 2 provide insufficient basis to make inferences about the impact of farm machine use on pesticide expenditure, since the simple comparison of mean differences fails to account for confounding factors such as observed (e.g., age, gender, household size, farm size) and unobserved characteristics (e.g., farmer’s motivation for farm machine use, farmers’ managerial skills).

## 4. Empirical Results

The full information maximum likelihood (FIML) method, a more efficient and consistent approach, is employed to estimate the selection and outcome equations simultaneously. The results of the determinants of farm machine use and the determinants of pesticide expenditure for farm machine users and nonusers are presented in Table 3. In the next section, we first discuss the determinants of farm machine use, which are estimated using the selection equation. Then, we discuss the determinants of pesticide expenditure. Finally, the estimates of ATT and ATU are presented and discussed.

### 4.1. Determinants of Farm Machine Use

The gender variable is negative and statistically significant, suggesting the female household heads are more likely to use farm machines relative to the male household heads, a finding that is consistent with Ma et al. [19]. The coefficient of off-farm work variable is positive and statistically significant, suggesting that participation in off-farm work increases the probability of farm machine use. This finding is consistent with the income-effects of off-farm work because income earned from off-farm work activities enables farm households to purchase farm machines or farm machine services. The risk preference variable appears to have a negative and statistically significant impact on farm machine use. One of the plausible reasons is that spraying pesticides using the farm machines can reduce the risk of pesticide exposure, which is preferred by the risk-averse farmers. The positive and statistically significant coefficient of transportation condition variable suggests that better transportation conditions increase the probability of using farm machines by smallholder maize farmers. Because purchasing farm machinery services is one of the primary requirements for having access to machines, the better conditions of transportation are essential to access the farm machinery services [37,38].

Farm size appears to have a positive and statistically significant impact on the probability of farm machine use, a finding that is echoed by Ma et al. [19]. Cultivating large farm size usually requires more labor endowments in order to maintain or enhance agricultural productivity, while farm machine use has the potential to substitute farm labors. The coefficient of extension contact variable is positive and significantly different from zero, indicating that farmers with contact with extension services are more likely to use farm machines. This finding is consistent with the finding of Abdulai [39], who found that the extension service tends to be a major source of information on technological improvement. The extension attitude variable, a proxy for farmers’ attitude to the extension service provided by the local government, has a negative and statistically significant impact on the farm households’ decisions to use farm machines. One possible reason for this could be that the agricultural extension service provided by the local government is insufficient and not related to farm machines. The project variable is also negative and significantly different from zero. This finding suggests that farmers who reside in a village which has implemented the projects related to environmental improvement are less likely to spray pesticides by using farm machines. These findings highlight the fact that the environment improvement projects can help improve farmers’ perception of the impacts of pesticide use, which enable farmers to apply more environment-friendly practices such as farm machine use [40].

Compared with farmers in the Gansu province, farmers in the Henan province are less likely to use farm machines to apply pesticides, whereas the farmers in the Shandong province are more likely to use farm machines for pesticide application. These findings suggest the presence of location fixed-effects resulting from the discrepancies of geographical conditions and institutional arrangements that may affect farmers’ choices to use farm machines when applying pesticides in maize production.

### 4.2. Determinants of Pesticide Expenditure

The determinants of pesticide expenditure for farm machine users and nonusers are presented in the third and fourth columns of Table 3, respectively. Our results show that the transportation condition variable has significant and positive impacts on pesticide expenditure for both farm machine users and nonusers. The findings suggest that transportation condition is a critical determinant of higher pesticide expenditure because better transportation conditions enhance farmers’ access to the inputs market. The age variable tends to have a negative and significant impact on pesticide expenditure for farm machine users, indicating that the elderly household heads decrease the pesticide expenditure due to the poorer health levels and lack of relevant skills used for pesticide application. The negative and significant coefficient of the age variable for farm machine users suggests that male household heads spend less on pesticides. The farming experience variable, which is proxied by farming years of the household head, is positive and statistically significant in the pesticide expenditure specification for farm machine users but not statistically significant in the specification for nonusers. This finding indicates that an increase of one year in the farming year of the household heads tends to increase pesticide expenditure by 0.86 Yuan. The result echoes the finding by Denkyirah et al. [41], who found that the years of farming experience have a positive and significant impact on pesticide use frequency in Ghana. The risk preference variable shows a negative impact on pesticide use, although the effect is insignificant even at the 10% significance level. The finding is in line with the findings of Gong et al. [42] and Liu and Huang [43]. For example, Liu and Huang [43] found that the farmers who are more risk averse use more quantities of pesticides in the production of BT cotton in China.

The coefficient of the variable for farm size in Table 3 is negative and statistically significantly different from zero for farm machine nonusers, suggesting that an additional unit (i.e., mu in the present study) increase in farm size tends to reduce pesticide expenditure by 3.33 Yuan/mu. This finding is consistent with the findings of Ma et al. [44] and Wu et al. [45]. For example, Wu et al. [45] found that farm size has significant positive effects on fertilizer and pesticide use per hectare. One possible explanation is that farmers with large farm sizes typically have relatively better management skills and farming knowledge to use chemical inputs efficiently. The coefficient of the project variable for the farm machine nonusers is negative and statistically significant, suggesting that the environment improvement project tends to encourage farmers to reduce the expenditure of pesticide. With regard to the location variables, the results in Table 3 show that farmers located in the Henan and Shandong provinces are associated with lower pesticide expenditure, relative to their counterparts in Gansu (reference group). The significance of the regional variable indicates that there exist location fixed-effects (e.g., institution arrangement, rainfall, and soil quality) that may influence farmers’ pesticide use behaviors.

In the lower parts of Table 3, we present the likelihood ratio test for joint independence of the selection equation and outcome equations. The statistic of the likelihood ratio test is significantly different from zero, suggesting the selection and outcome equations are dependent. That is, the three equations should be estimated jointly using the ESR model. As indicated previously, the correlation coefficients of the error terms ($\rho_{\mu U}$ and $\rho_{\mu N}$) are both positive and statistically different from zero, suggesting the presence of self-selection arising from unobservable factors [32]. The findings confirm the validity of using the ESR model to estimate the impact of farm machine use on pesticide expenditure, and suggest that using farm machines to spray pesticides may not generate the same effect on the nonusers if they choose to use farm machines [27]. In addition, the positive sign for $\rho_{\mu U}$ suggests a negative selection bias, indicating that farmers with below-average expenditure on pesticides have a higher likelihood to use farm machines. It is significant to mention that the positive selection bias in the present study is quite plausible since farm machines are expected to enhance the efficiency of pesticides applying and decrease the expenditure on the pesticides.

### 4.3. Estimations of the Treatment Effects

The estimates for the ATT and ATU, which show the treatment effects of farm machine use on pesticide expenditure, are presented in Table 4. It is important to note here that these ATT and ATU estimates are systematically different from the mean differences presented in Table 2, because they account for selection bias arising from both observable and unobservable factors.

The results for ATT estimates reveal that farm machine use tends to decrease the total pesticide expenditure by 58.87% or 28.32 Yuan/mu. Table 4 also presents the ATU results estimated by Equation (8b); for farm machine nonusers, they would decrease the total pesticide expenditure by 32.66% (about 11.05 Yuan/mu) if they chose to use farm machines. Our findings are contrary to those of Takeshima et al. [46] who noted that the use of farm machines in Ghana is associated with more intensive use of seed and chemical input by using a cluster analysis method which cannot control for the influences resulted from the unobserved factors.

For the purpose of comparison, we also estimated the treatment effects of farm machine use on pesticide expenditure, using a propensity score matching approach. In particular, we employ the most commonly used techniques including nearest neighbor matching (NNM) and Kernel-based matching (KBM) techniques to estimate the ATT and ATU [29]. The results, which are presented in Table A1 in Appendix A, show that farm machine use has negative and significant effects on pesticide expenditure except for the ATT estimates. However, the PSM method, which could not take into account the unobservable factors such as farmers’ innate abilities and managerial skills, may produce biased estimates. For example, the results estimated by NNM and KBM techniques show a nonsignificant and downwards impact of farm machine use for the ATT estimates, relative to the results estimated by ESR model we observed in Table 4.

## 5. Conclusions

There is a growing body of literature that shows that farm machines boost farm production and agricultural efficiency, and promote sustainable and conservation agriculture. However, the impact of farm machine use on pesticide expenditure has not been previously analyzed. Whether and to what extent farm machine use influences the pesticide expenditure remains poorly understood. This paper investigated the factors that affect farmers’ decisions to use farm machines, and analyzed the impact of farm machine use on pesticide expenditure. The study used cross-sectional farm household level data of maize farmers collected from a randomly selected sample of 493 households from the Gansu, Henan, and Shandong provinces in 2017. Results from the farm machine use mean differences revealed statistical differences in pesticide expenditure between farm machine users and nonusers. However, the mean differences could not account for the effects of other confounding characteristics, which may provide a misleading conclusion. Given that farmers self-select themselves into farm machine users and nonusers, we employed an endogenous switching model to address the sample selection bias arising from both observed and unobserved factors.

The empirical findings generally showed that farm machine use has a significant impact on pesticide expenditure. In particular, farmers who used farm machines to spray pesticides tend to reduce the pesticide expenditure by 58.63%, while the farmers who did not use farm machines would decrease pesticide expenditure 33.42% if they chose to be farm machine users. These findings suggest that farm machines serve as an environment-friendly technology to reduce pesticide use and enhance sustainable agriculture. On the factors that influence a farmer’s decision to use farm machines, the results show that transportation condition and extension contact exert positive and statistically significant effects on the farm machine use decision.

The findings from this study do have policy implications for sustainable and conservation agriculture development by reducing pesticide expenditure through farm machine use. In particular, the positive and significant effects of transportation condition, extension contact, and off-farm work on farm machine use suggest that farm machine use in rural regions could be enhanced through strategies such as improving the transportation condition, establishing information channels such as extension services, and providing more off-farm work information. Furthermore, with the increasing feminization of agriculture due to the increased propensity for men to migrate to urban areas for better off-farm work opportunities than women, the genitive impact of household head gender on use of farm machines advocates for the incorporation of gender-specific interventions in developing farm mechanization programs.

A limitation of this study is that we have only considered the impacts of farm machine use on the total pesticide expenditure. Such effects may also exist due to the significant discrepancies of different types of pesticides such as herbicides and insecticide, which is a promising area for future studies examine with a disaggregated analysis. In addition, our analysis of the present study only focuses on the maize farmers and data collected from three provinces in China; studies focusing on other crops and other regions or other countries are necessary in order to obtain a better understanding of the heterogeneous impacts of farm machine use on pesticide expenditure in a broad context.

## Figures and Tables

**Table 1 ijerph-16-01808-t001:** Definition and summary statistics of the selected variables.

Variables	Definition	Mean	SD ^1^
Pesticide expenditure	Expense on pesticide (Yuan/mu) ^2^	25.750	28.700
Farm machine use	1 if a household uses farm machines for pesticides application, 0 otherwise	0.576	0.495
Age	Age of household head (year)	46.790	10.320
Gender	1 if household head is male, 0 otherwise	0.836	0.371
Education	Schooling year of household head (year)	6.779	2.760
Off-farm work	1 if household head participate in off-farm work, 0 otherwise	0.712	0.453
Farming experience	Years of household head farming (year)	25.44	10.54
Risk preference	Risk preference score (1–10) ^3^	2.586	1.865
Household size	Number of people residing in a household	4.552	1.447
Credit access	1 if farmer has access to credit, 0 otherwise	0.428	0.495
Transportation condition	1 if transportation from the village to the train/bus station is convenient, 0 otherwise	0.753	0.432
Farm size	Total farm size used to cultivate maize (mu)	3.514	2.956
Subsidy	1 if household receives the agricultural subsidy, 0 otherwise	0.221	0.415
Extension contact	1 if household receives extension service, 0 otherwise	0.203	0.403
Extension attitude	Attitude to the extension service provided by local government (1–5) ^4^	2.673	1.152
Project	1 if the village executes the environment improvement project, 0 otherwise	0.807	0.395
Gansu	1 if household resides in Gansu, 0 otherwise	0.327	0.469
Henan	1 if household resides in Henan, 0 otherwise	0.345	0.476
Shandong	1 if household resides in Shandong, 0 otherwise	0.329	0.470

Note: ^1^ SD = standard deviation; ^2^ Yuan is Chinese currency, 1 USD = 6.70 Yuan in 2017; 1 mu = 0.067 hectare; ^3^ response option for the risk preference was a self-reported score scaling from 1 very risk-averse to 10 very risk-taking; ^4^ response option for the extension attitude was: 5 very useful, 4 useful, 3 a little useful, 2 useless, and 1 very useless.

**Table 2 ijerph-16-01808-t002:** The mean differences in characteristics between farm machine users and nonusers.

Variables	Users	Nonusers	Diff.	*t*-Value
Pesticide expenditure	19.788 (21.077)	33.856 (35.058)	−14.067 ***	−5.538
Age	46.246 (9.929)	47.522 (10.816)	−1.275	−1.356
Gender	0.796 (0.796)	0.890 (0.890)	−0.094 ***	−2.805
Education	6.673 (2.541)	6.923 (3.034)	−0.251	−0.997
Off-farm work	0.778 (0.416)	0.622 (0.486)	0.156 ***	3.832
Farming experience	25.331 (10.111)	25.584 (11.123)	−0.253	−0.263
Risk preference	2.127 (1.612)	3.211 (2.003)	−1.084 ***	−6.650
Household size	4.521 (1.569)	4.593 (1.264)	−0.072	−0.547
Credit access	0.317 (0.466)	0.579 (0.495)	−0.262 ***	−6.009
Transportation condition	0.863 (0.345)	0.603 (0.490)	0.260 ***	6.905
Farm size	4.116 (3.441)	2.696 (1.843)	1.421 ***	5.422
Subsidy	0.081 (0.273)	0.411 (0.493)	−0.330 ***	−9.487
Extension contact	0.201 (0.401)	0.206 (0.405)	−0.005	−0.137
Extension attitude	2.289 (1.009)	3.196 (1.139)	−0.907 ***	−9.380
Project	0.782 (0.414)	0.842 (0.366)	−0.060 *	−1.682
Gansu	0.131 (0.337)	0.593 (0.492)	−0.463 ***	−12.386
Henan	0.299 (0.459)	0.407 (0.492)	−0.107 **	−2.490
Shandong	0.570 (0.496)	0.000 (0.000)	0.570 ***	16.625

Note: Standard deviation in parentheses; * *p* < 0.1, ** *p* < 0.05, *** *p* < 0.01.

**Table 3 ijerph-16-01808-t003:** Determinants of farm machine use and determinants of pesticide expenditure.

Variables	Selection	Pesticide Expenditure	
		Users	Nonusers
Age	−0.046 (0.032)	−0.902 (0.325) ***	0.132 (0.531)
Gender	−0.639 (0.223) ***	−12.466 (3.989) ***	−1.150 (7.853)
Education	−0.011 (0.035)	−0.322 (0.588)	−1.026 (0.814)
Off-farm work	0.319 (0.177) *	−1.444 (3.881)	1.777 (4.506)
Farming experience	0.039 (0.032)	0.857 (0.316) ***	0.424 (0.470)
Risk preference	−0.125 (0.050) **	−0.238 (0.662)	−0.500 (1.079)
Household size	0.082 (0.066)	−0.309 (0.514)	0.991 (1.962)
Credit access	−0.232 (0.193)	3.045 (2.243)	2.869 (6.358)
Transportation condition	0.628 (0.192) ***	9.329 (2.458) ***	12.919 (4.894) ***
Farm size	0.099 (0.033) ***	−0.347 (0.291)	−3.327 (1.078) ***
Subsidy	−0.413 (0.233) *	10.489 (6.888)	−4.728 (7.064)
Extension contact	0.386 (0.222) *	−5.240 (3.420)	−1.277 (6.717)
Extension attitude	−0.233 (0.079) ***	0.251 (1.469)	−1.010 (2.523)
Project	−0.619 (0.217) ***	−6.219 (3.971)	−20.540 (6.676) ***
Henan	−0.595 (0.298) **	−11.463 (4.758) **	−38.365 (7.615) ***
Shandong	6.388 (0.567) ***	−8.787 (4.572) *	
IV	2.376 (0.697) ***		
Constant	1.067 (0.867)	59.962 (11.964) ***	63.102 (21.863) ***
$Ln\sigma_{\mu U}$		2.871 (0.128) ***	
$\rho_{\mu U}$		0.056 (0.068)	
$Ln\sigma_{\mu N}$			3.351 (0.091) ***
$\rho_{\mu N}$			0.398 (0.135) ***
LR test of indep. eqns.	$\chi^{2}\left(2\right)=9.31{\;}^{**}$		
Log-likelihood	−2361.866		
Observation	493	493	493

Note: Standard errors in parentheses; * *p* < 0.1, ** *p* < 0.05, *** *p* < 0.01; The reference region is Gansu; Due to all the samples in Shandong use farm machines to spray pesticide, therefore the regime 0 will exclude the dummy variable of Shandong.

**Table 4 ijerph-16-01808-t004:** Impact of farm machine use on pesticide expenditure: endogenous switching regression (ESR) model estimation.

Variables	Category	Average Expected Expenditure (Yuan/mu)		Treatment Effects	*t*-Value	Change (%)
		Users	Nonusers			
Pesticide expenditure	ATT	19.788	48.107	−28.319 ***	−25.497	58.87
	ATU	22.780	33.827	−11.047 ***	−11.651	32.66

Note: * *p* < 0.1, ** *p* < 0.05, *** *p* < 0.01.

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
