# Peer review of "Farm Machine Use and Pesticide Expenditure in Maize Production: Health and Environment Implications"

_ijerph, 2019, doi:10.3390/ijerph16101808_

Round 1
Reviewer 1 Report
The reviewer appreciates the research effort. The copy received for reviewed is incomplete (I cannot believe that incomplete citations and missing files were left purposely). Some comments:
Abstract lines 17-18 suggested language: " In this study, we are trying to explore whether farm machine use enables to help reduces pesticide expenditure by analyzing farm household survey data collected ... "
Line 44- "which targeted safety food safety and conservation production"
Line 71- farm machine use is not randomly distributed (it seems that if it is self-selection it may be a random distribution),
Line 81- and conversation? Not sure what “conversation agriculture” means. Is it supposed to be “sustainable and conventional agriculture”?
Line 176- citation is incomplete: and Ma et al. (2018c) and xxx???
Line 178- misspelled- randomised is randomized
Lines 197, 201, 211, 228 and 330 show Error! Reference source not found.
Line 250- which is more adorable? Is it amenable to the risk aversion farmers
Line 353- Table A1 (there is no Table A1 in my file) in the Appendix
Line 403-404- incomplete statement:
Please add: This research received no external funding or This research was funded by [name of funder] grant number [xxx] And The APC was funded by [XXX].
Author Response
Response to Reviewer #1 We very much appreciate your constructive comments and have tried as best as we can to respond to each of them directly. The changes we have made in response to the points are as follows:
Comment 1 The reviewer suggests that we use the suggested language: "In this study, we are trying to explore whether farm machine use enables to help reduces pesticide expenditure by analyzing farm household survey data collected ... " in the Abstract section.
Response to Comment 1 In line with your suggestion, we have now deleted “enables to help” in the relevant sentence.
Comment 2 Line 44: The reviewer suggests that we replace “safety food” with “food safety”.
Response to Comment 2 Your suggestion has been fully taken.
Comment 3 Line 71: The reviewer points out that farm machine use seems that if it is self-selection it may be a random distribution.
Response to Comment 3 In response to your concerns, we have now re-written the relevant sentence as “In addition, farmers themselves decide whether or not to use farm machine for pesticide application (i.e. a self-selection process)”.
Comment 4 Line 81: The reviewer wants to know the “conversation agriculture” means.
Response to Comment 4 It should be “conservation agriculture” rather than “conversation agriculture”. We have now corrected it. We really appreciate your professional and detailed comments.
Comment 5 Line 176: The reviewer points out the citation is incomplete.
Response to Comment 5 In response to your suggestion, we have now re-written the relevant sentence (please see Page 5).
Comment 6 Line 178: The reviewer points out that we misspelled the “randomized” to “randomised”.
Response to Comment 6 “Randomised” is the British spelling, and the “randomized” is the preferred North American spelling. In response to your concerns, we have now used the spellings that you have suggested.
Comment 7 Lines 197, 201, 211, 228 and 330: The reviewer points out that the sentences show Error! Reference source not found.
Response to Comment 7 In response to your suggestion, we have now corrected the cross-reference errors to make it more readable.
Comment 8 Line 250: The reviewer suggests that this sentence is not clear. Response to Comment 8 In response to your suggestion, we have now re-written the relevant sentence to make it more readable.
Comment 9 Line 353: The reviewer points out that there is no Table A1 in the Appendix.
Response to Comment 9 We have now carefully checked our manuscript and labelled Table A1 (Please see Page 10 and Page 11)
Comment 10 Line 403-404: The reviewer suggests that we completed the funding statement.
Response to Comment 10 In line with your suggestion, we have now added the detailed funding information.

Reviewer 2 Report
Title: Farm machine use and pesticide expenditure in maize production: health and environment implications.
This paper is well done and the study is very interesting because it takes into consideration an important problem like pesticides uses and sustainable agriculture. I agree with the authors when they say that it is very important to reduce pesticide amount and expenditure because of human and environmental health risks associated with their use.
The introduction provides sufficient background and includes relevant references in my opinion. The research design seems to be appropriate and methods are adequately described. Results are clearly presented but, because I’m not a statistician, the reported conclusion are not so immediate for me.
However some clarifications are necessary.Please, check and replace “Error! Reference source not found” in all paper.
Line 35: please, move to previous line “Figure A1 in the Appendix”.
Line 41: please, add the paper number when you write Lai (2017). It becomes: Lai (2017) [3]. Please, do it for all cited articles in the paper.
In the Equation 1, Xi refers to several farmer characteristics like age, gender, educational level, etc. . All these variables were considered in the function Yi. Also they were considered in functions 4a and 4b in Zi : how can you explain this overlapping? Did you consider these variables twice?
Line 176: please, replace “xxx” with a reference or delete it.
Lines 365-366: how do you obtain 58.63% and 33.42%? Where did you report these results? Not in Table 2.
It would be interesting to know results of your study if you could consider two types of pesticides: herbicides and insecticides. Do you think to continue the study?
Author Response
Response to Reviewer #2
We very much appreciate your careful reading of our manuscript and detailed comments and suggestions. They have helped greatly improve the quality of our paper. We have tried as best as we can to respond to each of them directly. The changes we have made in response to the points are as follows:
Comment 1 The reviewer suggests we check and replace “Error! Reference source not found” in all paper.
Response to Comment 1 We have now checked the errors and addressed them thoroughly.
Comment 2 Line 35: The reviewer suggests we move the term “Figure A1 in the Appendix” to the previous line.
Response to Comment 2 Your suggestion has been fully taken.
Comment 3 Line 41: please, add the paper number when you write Lai (2017). It becomes: Lai (2017) [3]. Please, do it for all cited articles in the paper.
Response to Comment 3 Your suggestions have been fully taken.
Comment 4 In the Equation 1, Xi refers to several farmer characteristics like age, gender, educational level, etc. . All these variables were considered in the function Yi. Also they were considered in functions 4a and 4b in Zi: how can you explain this overlapping? Did you consider these variables twice?
Response to Comment 4 We appreciate your concerns on this particular issue. In the Endogenous switching regression framework, the variables ?? in Equation (3) and ?? in Equation (1) are allowed to overlap (Lokshin and Sajaia 2004; Ma and Abdulai 2016). However, proper identification requires at least one variable in ?? that does not appear in ??. Thus, the selection equation is estimated using the same variables in the outcome equation in addition to at least an identifying instrument.
Comment 5 Line 176: The reviewer suggests we replace “xxx” with a reference or delete it.
Response to Comment 5 In response to your suggestion, we have now deleted it.
Comment 6 Lines 365-366: The reviewer wants to know how to obtain 58.63% and 33.42%? Where did we report these results?
Response to Comment 6 The change percentage values are calculated by ????????? ??????? ??????? ???????? ??????????? ??? ?ℎ? ???????? , and the results are presented in Table 4.
Comment 7 It would be interesting to know results of your study if you could consider two types of pesticides: herbicides and insecticides. Do you think to continue the study?
Response to Comment 7 That is a genius concern. It is definitely be an interesting area to consider herbicides and insecticides, which would be a future direction for us to explore.
References Lokshin, M. and Z. Sajaia. (2004). Maximum likelihood estimation of endogenous switching regression models. The Stata Journal, 4(3): 282–289. Ma, W. and A. Abdulai. (2016). Does cooperative membership improve household welfare? Evidence from apple farmers in China. Food Policy, 58: 94–102.

Reviewer 3 Report
1. Several of the references are missing (eg. line 198).
2. Please write in detail about other regression models that are used.
3. Compare and contrast PSM with ESR based approach in more detail.
Author Response
Response to Reviewer #3 We very much appreciate your constructive comments and have tried as best as we can to respond to each of them directly. The changes we have made in response to the points are as follows:
Comment 1 Several of the references are missing (eg. line 198).
Response to Comment 1 In response to your concerns, we have now carefully corrected the reference issues that appeared in the early version.
Comment 2 Please write in detail about other regression models that are used.
Response to Comment 2 In line with your suggestion, we have now added more discussions about other regression model we have used (please see Page 3)
Comment 3 Compare and contrast PSM with ESR based approach in more detail.
Response to Comment 3 In line with your suggestion, we have now added a comparison between the ESR model and the PSM model (see Page 3 and Page 10).
